# Goal management training and psychoeducation / mindfulness for treatment of executive dysfunction in Parkinson's disease: A feasibility pilot trial

Ariane Giguère-Rancourt[1]*, Marika Plourde[1], Eva Racine[1], Marianne Couture[1], Mélanie Langlois[2,3], Nicolas Dupré[2,3], Martine Simard[1]

1 School of Psychology, Laval University, Quebec City, Canada, 2 Department of Medicine, Faculty of Medicine, Laval University, Quebec City, Canada, 3 Axe Neurosciences du CHU de Québec, Université Laval, Québec, Quebec City, Canada

* ariane.giguere-rancourt.1@ulaval.ca, ariane.grancourt@gmail.com

## Abstract

### Introduction

As there is currently no pharmacological treatment for Parkinson's Disease Mild Cognitive Impairment (PD-MCI) with executive dysfunctions, specific cognitive interventions must be investigated. Most previous studies have tested bottom-up cognitive training programs but have not shown very good results.

### Objectives

The aim of this study was to test ease of implementation, differential safety and preliminary efficacy of two top-down (strategy-learning) home-based, individualized, cognitive interventions: Goal Management Training (GMT), adapted for PD-MCI (Adapted-GMT), and a psychoeducation program combined with mindfulness exercises (PSYCH-Mind).

### Methods

This was a single-blind block-randomized between-group comparative study. Twelve PD-MCI with mild executive dysfunctions were divided in four blocks and randomly assigned to any of the two interventions. The participants were included if they had PD-MCI diagnosis (no dementia), with stabilized medication. Both groups (Adapted-GMT and PSYCH-mind) received five intervention sessions each lasting 60–90 minutes for five weeks. Measures were collected at baseline, mid-point, one-week, four-week and 12-week follow-ups. Executive functions were assessed with the Dysexecutive questionnaire (DEX) and the Zoo Map Test (ZMT). Quality of life (QoL) and psychiatric symptoms were also evaluated. Repeated measures ANCOVAs (mixed linear analysis) were applied to all outcomes.

### Results

There was one drop out, and both interventions were feasible and acceptable. Despite the small sample size limiting statistical power, patients of both groups significantly improved

**Data Availability Statement:** Data availability: The data that support the findings of this study are

available from the corresponding author, AGR, upon reasonable request. You can also contact Laval University data research manager at the following email address: gdr@ulaval.ca Data can be accessed at the following page (please send an access request): Giguère-Rancourt, Ariane, 2020, "Cognitive Training in Parkinson's disease", https://doi.org/10.5683/SP2/0LMBRX, Scholars Portal Dataverse, V1, UNF:6:RAQT1DXYblrlo +gpYplHyA== [fileUNF].

**Funding:** Funding details: AGR was supported by a doctoral training award from Laval University's Fonds Facultaire d'Enseignement et de Recherche (FFER). The funders had no role in study design, data collection and analysis, decision to publish, or preparation of the manuscript.

**Competing interests:** The authors have declared that no competing interests exist.

executive functions per the DEX-patient (Time: $F_{(4,36)} = 2.96$, $p = 0.033$, CI95%: 10.75–15.23) and DEX-caregiver scores (Time: $F_{(4,36)} = 6.02$, $p = 0.017$, CI95%: 9.63–17.23). Both groups significantly made fewer errors between measurement times on the ZMT (Time: $F_{(3,36)} = 16.66$, $p = 0.001$, CI95%: 1.07–2.93). However, QoL significantly increased only in PSYCH-Mind patients at four-week follow-up (interaction Time*Group: $F_{(4,36)} = 5.31$, $p = 0.002$, CI95%: 15.33–25.61).

## Conclusion

Both interventions were easily implemented and proved to be safe. Because both interventions are arguably cost-effective, these pilot findings, although promising, need to be replicated in large samples.

## ClinicalTrials.gov Identifier

NCT04636541.

## Introduction

Mild Cognitive Impairment (MCI) affects approximately 30% of people with Parkinson's disease (PD) [1]. Nearly 40% of PD-MCI patients report difficulties with executive functions [2, 3].

Executive functions involve integrative processes supporting goal-directed behaviors [2]. For example, dysexecutive patients experience difficulties to plan and execute complex activities requiring multiple steps, such as preparing meals or maintaining attention to tasks without being distracted. Degeneration of frontal-striatal loops in PD-MCI might explain behavior disorganization [3, 4]. Medications for motor symptoms might improve some cognitive functions but deteriorate others in *de novo* PD and PD-MCI [5–8]. Therefore, non-pharmacological approaches, such as cognitive training, shall be considered as possible treatment for cognitive impairment.

Studies assessing the impact of cognitive training on executive functions in PD-MCI are still scarce [9]. Our group recently conducted a systematic review on trials of cognitive interventions administered to PD patients with and without MCI [9]. Only 13 studies met the inclusion criteria of the review. Cognitive training programs described in these studies involved between 10 and 16 sessions, each lasting from 30 to 90 minutes, and were mostly *drill-and-practice* or *bottom-up* training with repetition of paper-and-pencil tasks and computer tasks to improve executive functions as well as other cognitive functions. The review reported that in these trials, PD-MCI, compared with controls, only improved performance in 6.9% of cognitive measures after mostly 4 to 6 weeks of cognitive training. However, PD-MCI who received only cognitive training (as opposed of those who received other kinds of cognitive interventions) did not improve on executive functions when compared to controls [9]. The lack of efficacy in these trials to improve executive functions could be explained partly by methodological limitations and partly by the *bottom-up* training approach chosen so far.

Thus, the trials published to date present methodological limitations [9, 10]. First, cognitive intervention programs were generally not developed to deal with subjective cognitive complaints or objective executive deficits specific to participants; in other words, regardless of the MCI type of a particular patient, all participants received the exact same cognitive training to

improve the same cognitive functions. This method does not consider the heterogeneity of MCI in PD [1, 2, 11]. This could induce either a ceiling or a floor effect, depending on the presence of dysexecutive functions at baseline. This problem could be avoided by ensuring that PD-MCI involved in cognitive training of executive functions do in fact present with executive impairment at baseline. This limitation could partly explain the absence of effect or small effect sizes reported in previous studies [9]. Lack of assessment for neuropsychiatric symptoms is another methodological limitation as these symptoms are frequent in PD and could affect feasibility, safety, and efficacy of intervention [12, 13]. Furthermore, important constructs conceived as indicators of generalization, such as quality of life and subjective cognitive complaints made by patients and caregivers, have not been examined in prior research [9, 10]. Finally, one should question the *bottom-up* approach to improve executive functions adopted in previous trials. A bottom-up training approach makes it more difficult to transfer skills acquired in training session to day-to-day functioning since it focuses on the practice of specific tasks [14]. Most of the time, the efficacy of this kind of training is assessed on the same tests used to perform the training, thus making generalization of training questionable. On the contrary, a strategy-based or top-down cognitive training approach which consists in teaching different cognitive strategies applicable to many daily situations [14], has a greater potential for improving the daily functioning of dysexecutive PD patients. However, strategy-based approaches were rarely tested in PD-MCI patients.

The Goal Management Training (GMT) is a *strategy-based* or *top-down* cognitive training program [15–17] as opposed to *the drill-and practice* or *bottom-up* cognitive training used in prior studies in PD. The GMT, based on Duncan's theory of goal neglect [15], has been developed to improve executive functions through the management and prioritization of goals in the achievement of daily activities. To teach patients to better manage their daily goals, the GMT program includes, in each of the nine sessions, self-instruction strategies, self-monitoring exercises, cognitive training techniques such as goal definition or task splitting, psychoeducation on cognitive processes, mindfulness and other exercises to practice newly acquired knowledge between sessions. In a systematic review and meta-analysis assessing the effectiveness of GMT [17], the studies included patients with various neurological conditions affecting frontal lobe functioning such as acquired traumatic brain injury, *spina bifida*, attention deficit and hyperactivity disorder, subjective cognitive complaints, and multiple sclerosis [17]. The GMT has been shown to increase those patients' awareness of deficits and improve cognitive control in goal-directed behaviors, as well as other executive functions measured by neuropsychological tests and questionnaires [17]. However, the GMT has never been studied in PD-MCI up to date. Although patients who received the GMT so far present with neurological disorders other than PD, they exhibit dysexecutive functions caused by impaired frontal cortex networks, as do PD-MCI patients with executive functions impairment. Thus, GMT might be suitable for PD-MCI patients with executive dysfunction. However, the original GMT was developed in English, was not yet available in French, and is a nine-week program administered in group sessions of 90 to 120-minutes [16]. This program might be too long, tiresome, and difficult to follow for PD patients. In addition, the obligation to leave home to attend group sessions might prove difficult if not impossible for patients with mobility issues as those with PD. All these factors could affect clinical trials feasibility in vulnerable patients with PD-MCI. Therefore, our team first translated in French and adapted the GMT to be administered at home for a PD-MCI patient in a case study [18]. Up to four weeks after the end of Adapted-GMT, the participant had not missed any training sessions, and showed a significant decrease in executive complaints according to the DEX score. Furthermore, the participant did not register any increase in neuropsychiatric symptoms, suggesting that the Adapted-

GMT is acceptable and safe. However, this study does not report any data comparing the effects of Adapted-GMT with other treatment and only investigated one patient.

The present pilot study thus aimed at testing feasibility, safety and preliminary efficacy of the Adapted-GMT compared with another top-down treatment less systematically studied in the past. Many clinical guidelines include general recommendations about giving information to PD patients and family [19, 20]. However, few standardized psychoeducation interventions are available, and they do not include information on cognitive decline [21–23]. Another less studied treatment in PD, the *Mindfulness Based Stress Reduction* program includes formal meditative exercises to develop non-judgmental attention to experiences in the present moment [24, 25]. In elderly patients with MCI unrelated to PD, mindfulness interventions have shown positive effects on cognitive functioning, including attention, executive functions, and memory [26, 27]. In PD, previous trials mostly investigated the efficacy of mindfulness to reduce psychological symptoms, but none of them studied its impact on cognition. In addition, a review of the literature on the effect of mindfulness in PD shows that methods and results are inconsistent and that more studies are needed in PD patients [24].

In summary, Adapted-GMT, psychoeducation, and mindfulness are top-down interventions that have never been investigated to improve executive functions in PD. Nonetheless, they are potentially and theoretically interesting options for treatment of executive dysfunctions in patients with PD-MCI, partly because they are less likely to cause adverse effects than the pharmacological interventions. However, before testing the effectiveness of these interventions with large patient samples, one must first verify their feasibility, safety and preliminary efficacy, especially when offered at home, and in the presence of a caregiver. Therefore, the present pilot study shall investigate these aspects, in a small sample of PD-MCI.

## Objectives and hypotheses

The main objective of this study was thus to compare feasibility, safety and preliminary efficacy of two interventions: Adapted-GMT versus Psychoeducation combined with Mindfulness exercises, in two groups of PD-MCI patients experiencing executive dysfunctions.

Hypotheses are: 1) Both interventions will prove to be feasible, safe, and acceptable by patients and caregivers; 2) Only Adapted-GMT will improve executive functioning, as measured by neuropsychological tests; 3) Both Adapted-GMT and Psychoeducation- Mindfulness (PSYCH-Mind) will improve subjective executive functioning, as shown with questionnaires; 4) Both interventions will maintain quality of life as measured with the Parkinson Disease Questionnaire-39 items (PDQ-39) and global cognition as measured with the Dementia Rating Scale 2$^{nd}$ edition (DRS-II), compared to baseline (BL).

## Methods

### Participants

Participants were recruited from the *Parkinson Québec Network* through announcements in gatherings and newsletters. Participants were also recruited at the Movement Disorder Clinic of *CHU-Hôpital Enfant-Jésus*, in Quebec City, as described in previous paper [18]. Flowchart of recruitment is displayed in Fig 1.

Participants had to meet the following inclusion criteria: 1) the UK Research Brain Bank diagnostic criteria for PD [28]; and 2) the Movement Disorder Society Task Force diagnostic criteria for PD-MCI [1]. Single and multiple-domain MCI were both included, only if executive functions were significantly impaired, with scores between -1 SD and -2 SD on executive function tests, according to age- and education-adjusted norms; 3) MoCA scores between 21 and 27; 4) Anti-Parkinson medication stable at screening since at least two months; and 5) All

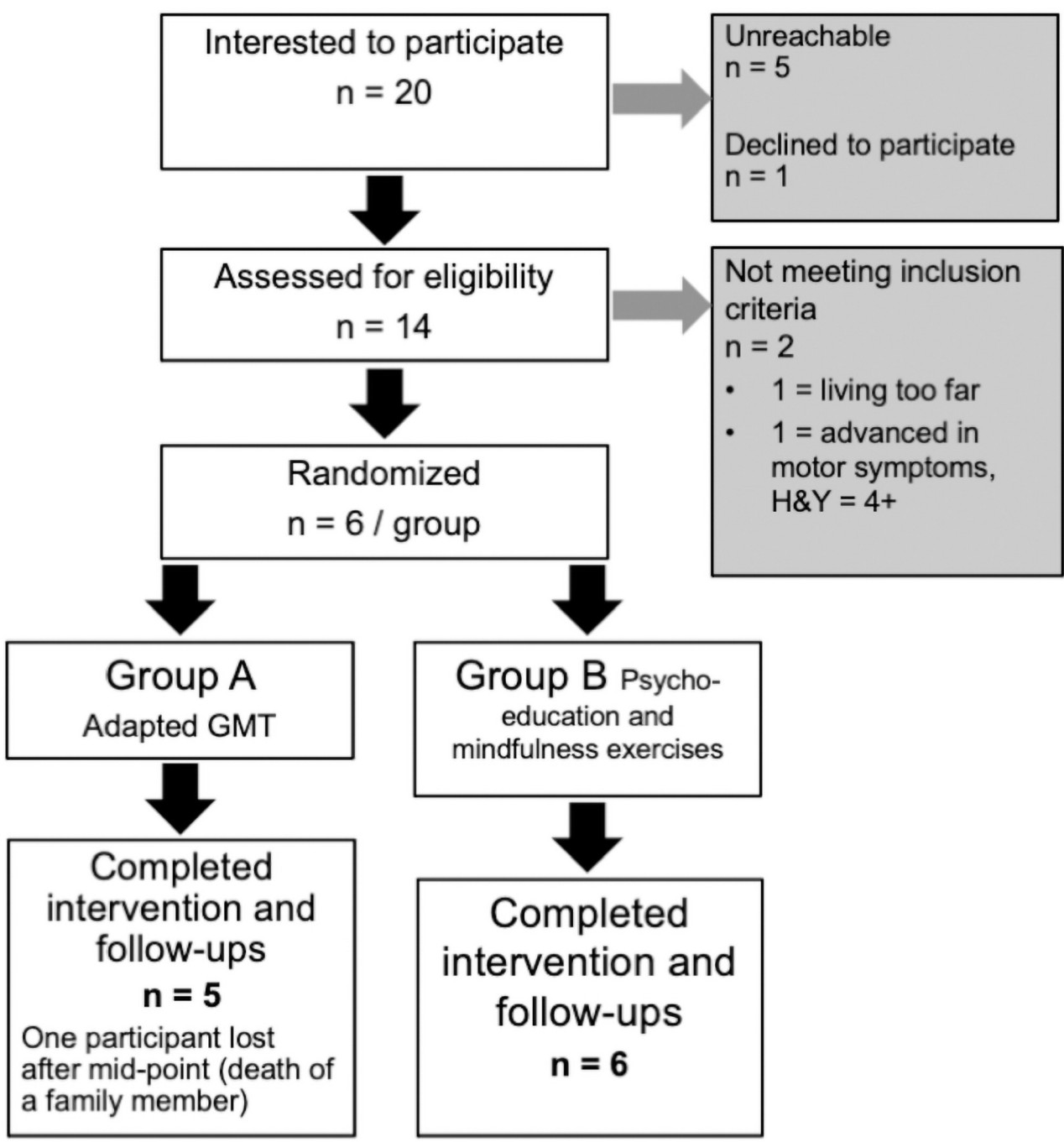

**Fig 1. Flow diagram of the final sample.**

other medications, including psychotropics, stable for at least three months. Exclusion criteria were: 1) Participants with PD meeting the clinical diagnostic criteria for dementia (PDD) [29]; 2) Patients with concomitant other neurological or psychiatric disorders and 3) patients with PD-MCI unstable on anti-Parkinson medication at screening or stable since less than 2 months on these medications; and 4) PD patients unstable or stable since less than three months on all other medications.

## Ethical considerations

This study was approved by the Ethics and Research Committee of *CHU de Québec–Université Laval*. Before entering into the study, patients and caregivers were fully informed about the project and signed an informed consent. All nominative data were kept confidential by coding of documents. The Ethics and Research Committee of *CHU de Québec–Université Laval* required the dataset to be kept in encrypted files, with the access protected by a password only known by the doctoral student (A. Giguère-Rancourt) and the laboratory director (Dr. Simard). Data of this study are available from the corresponding author only upon reasonable request.

## Research design, randomization, allocation procedure and course of the study

The present research was a single-blind block-randomized between-group comparative study (see Fig 2 for study protocol).

Clinical trial registration number is NCT04636541. Doctoral dissertation research projects at Laval University, Canada, only require the approval of the Ph.D. Candidate's advisory committee followed by the approval of the local ethics committee to start the study. The various steps of doctoral training having very tight deadlines, and the project not being funded by a national agency, the research started immediately after the ethical approval. This is the reason why the study protocol was not registered before it started. The authors confirm that all ongoing and related trials for this intervention are registered.

Three randomization blocks, each including four participants, were created during recruitment. Participants of the same block were assessed and received interventions over the same time period. After screening evaluation, participants of each block were randomly assigned to either group A or B, described below. Two participants of each block were thus in Group A (GMT) whereas two other participants were in Group B (PSYCH-Mind). Participants then received baseline evaluation. After the baseline evaluation, participants received five intervention sessions, each lasting 60–90 minutes, for five weeks. Evaluations and interventions took place at participants home, in presence of caregivers. All participants and caregivers were blinded to group allocation. Follow-up assessments were administered by the investigator administering the interventions but were scored by an independent rater blind to previous performances. The same investigator administered all interventions and evaluations for standardization of administration of tests and interventions and to control for clinician characteristics. All questionnaires were administered halfway through the intervention (week 3) and one, four and 12 weeks after the end of the intervention. All questionnaires and tests were rated after the administration and results were entered in the database by independent raters, not informed of the group allocation.

**Group A: Goal Management Training (GMT).** GMT modules [16] were adapted for French-speaking patients with PD-MCI. Each session was reduced from nine 90-120-minute sessions (original GMT) to five 60-90-minute sessions, one session per week, to avoid fatigue as much as possible. As for original GMT, participants were given exercises between sessions (mindfulness exercises and metacognitive reflections). All participants reported they practiced the tasks between sessions (between 30 and 150 minutes per week). In original-GMT, some information is repeated several times, but not in Adapted-GMT, to reduce sessions duration and thus avoid fatigue and attrition. Exercises demanding motor dexterity, such as card distribution, were removed because of PD motor impairment. Adapted-GMT included information on PD-MCI and executive dysfunction unlike the original GMT. In addition, Adapted-GMT modules were administered individually with an iPad, as opposed to a power-point group

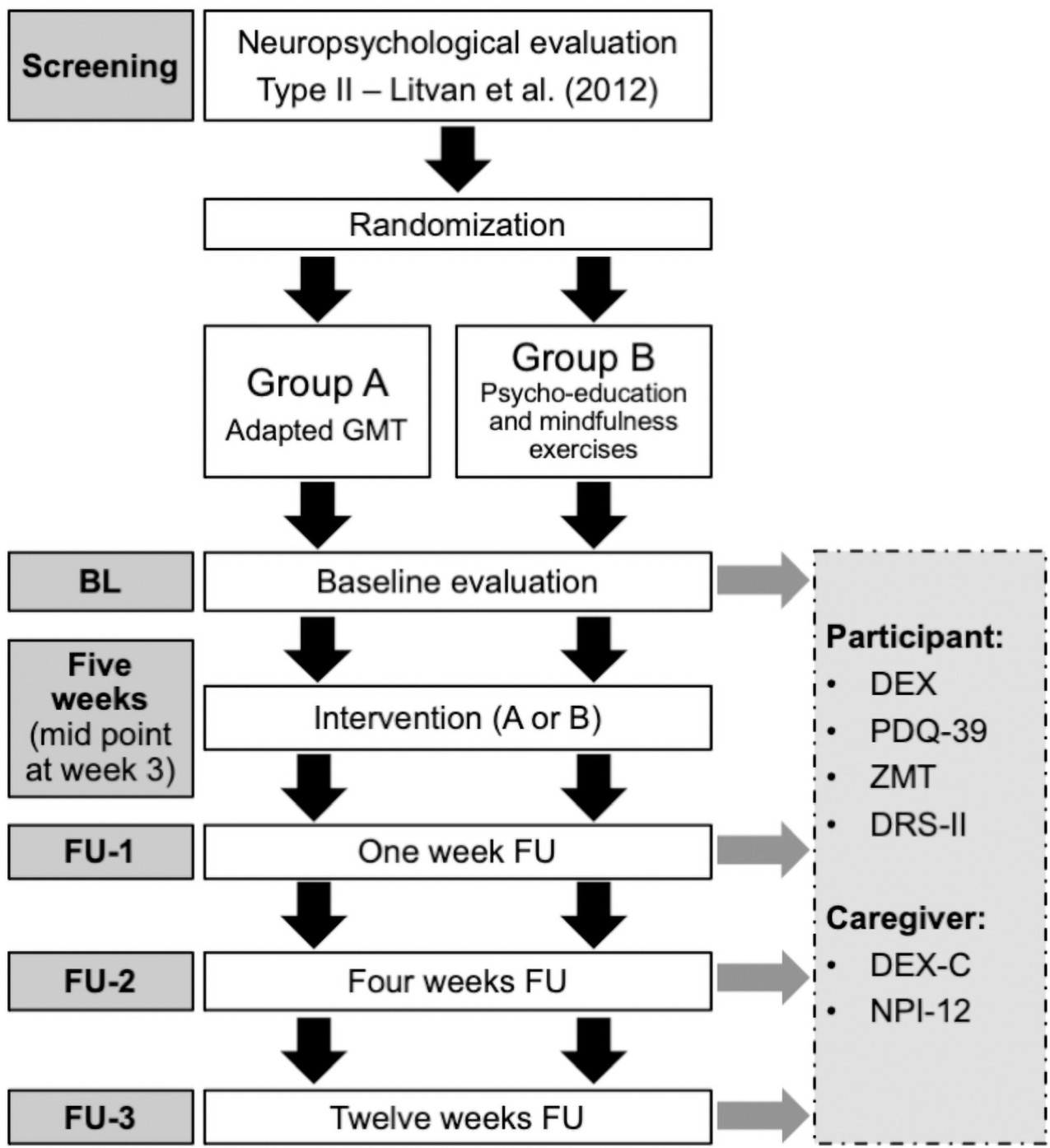

**Fig 2. Study protocol.**

presentation in original-GMT. A workbook was handed to participants, as in previous studies. The original GMT material, including the version translated in French for the present project, is now available for sale on Baycrest Centre website under the name Goal Management Training® Products [30]. Dr. Brian Levine and Baycrest Center are the owners of the copyrights.

Prior to the development of the present study, Dr. Levine and Baycrest Centre granted special permission to the authors to translate into French, adapt and use the GMT material for patients with PD-MCI included in this study.

**Group B: Psychoeducation with mindfulness (PSYCH-Mind) exercises.** Five modules were designed as a discussion with patients and caregivers about various PD symptoms: module I- brain and motor symptoms; module II- autonomic symptoms; module III- psychological symptoms; module IV- brain and cognition; and module V- cognitive impairments in PD. Patients were handed the information book at the beginning of the study. The objective was to improve their understanding of their condition and to discuss other components that could affect cognitive abilities. After the 40-to-60-minute informative part, mindfulness exercises were offered for 20–30 minutes per session. Participants were not invited to practice exercises between sessions, but 3/6 participants reported they did.

The same mindfulness exercises were practiced with all participants of Groups A and B. There were four exercises: mental visualization of 'eating a grape' in mindfulness (as an introduction); body scan; long breathing (12 minutes) and short breathing (three minutes).

## Evaluation

**Feasibility measures.** Although some authors offer elements to consider when researchers develop feasibility studies [31, 32], there are currently no recognized standards for such studies. Some authors [31, 32] suggest several key elements that may be studied as feasibility issues depending on the population targeted by a treatment, the treatment itself, and context. Among the proposed focuses by these authors, the present study examined: 1) acceptability, using the following outcomes: satisfaction, intent to continue to use, perceived appropriateness; 2) demand, using the following outcomes: expressed interest or intention to use by patients); 3) implementation, using the following outcomes: degree of execution, success or failure of execution; 4) practicality, with the following outcomes: amount and type of resources needed to implement, i.e. calculation made by analyzing the number of hours worked by the therapist for each participant, including transport, pre-intervention assessment, intervention and correction of questionnaires and protocols, and finally, safety issues on target participants and ability of participants to carry out intervention and activities as measured with attrition rate; 5) limited or preliminary efficacy testing with the following outcomes: intended effects of program on key variables, effect-size estimation, maintenance of changes from initial change. A last element of feasibility that will be considered in the present paper is 6) adaptation, and this final point will be addressed in the discussion of the article by attempting to answer these two questions: 1) How do present GMT results obtained by PD-MCI compare with results obtained previously by participants with other neurological conditions? How do the adapted GMT and PSYCH-Mind results compare with results of previous cognitive training studies in PD-MCI?.

**Safety measures.** To assess safety and tolerability, attendance, medication change (Levodopa Equivalent Dosage or LED and other medications if appropriate), and fatigue, using a Likert scale scoring from 0 to 10; 0 = none, 10 = extreme fatigue, were recorded at each session (means +/- SD are reported). For the fatigue scale, 0–1 was associated with no fatigue; from 2 to 4 was associated with mild fatigue, 6–7 to moderate fatigue, 8–9 to severe fatigue and 10 was associated with intolerable fatigue.

**Screening measures administered before randomization and baseline.** Neuropsychological evaluation–Type II [1] was administered at screening to determine MCI subtype (see Table 1).

**Efficacy measures administered at baseline, during training and follow-ups.** All assessments and interventions were performed by a Senior Ph.D. Candidate under the supervision

**Table 1. Descriptive statistics of the final sample, screening evaluation results and descriptive of primary and secondary outcomes at baseline.**

| | Group A Mean (SD) N = 6 | Group B Mean (SD) N = 6 | *p* |
|---|---|---|---|
| **Age** | 71.00 (4.03) | 70.00 (5.09) | 0.77 |
| **Sex (*n* men / 6)** | 5 / 6 | 5 / 6 | N/A |
| **Education (years)** | 17.17 (4.32) | 16.00 (3.30) | 0.84 |
| **Occupation (*n* retired / 6)** | 4 | 4 | N/A |
| **Time since diagnostic (years)** | 10.33 (4.41) | 5.44 (3.31) | 0.006** |
| **H&Y stage** | 1.92 (0.58) | 1.67 (0.75) | 0.53 |
| Stage 1–1.5 (n / 6) | 2 | 3 | N/A |
| Stage 2–2.5 (n / 6) | 3 | 2 | N/A |
| Stage 3 (n / 6) | 1 | 1 | N/A |
| **LED (mg / day)** | 728.94 (297.36) | 637.32 (436.91) | 0.69 |
| **Comorbidities** | | | |
| Sleep problem treated with medication (n / 6) | 1 | 2 | N/A |
| Hypertension (n / 6) | 2 | 2 | N/A |
| Type II Diabetes (n / 6) | 0 | 1 | N/A |
| **Caregivers** | | | |
| N | 6 | 5 | N/A |
| Age | 66.33 (4.92) | 63.5 (8.21) | 0.49 |
| Sex (n women) | 6 | 5 | N/A |
| **Screening Neuropsychological Evaluation Type II** | | | |
| **Visuospatial skills** | | | |
| Visual puzzle WAIS-IV (Z score) [33] | 0.00 (0.55) | 0.22 (1.44) | 0.64 |
| BJLO (Z score) [34] | 0.50 (0.77) | 0.92 (0.58) | 0.52 |
| **Language** | | | |
| Lexical fluency, T-N-P (Z score) [35] | -0.07 (1.19) | 0.50 (0.68) | 0.34 |
| Category fluency, Animals (Z score) [35] | -0.34 (1.17) | -0.29 (0.48) | 0.92 |
| NAB Screening-Naming (Z score) [36] | 0.60 (0.41) | -0.05 (1.35) | 0.22 |
| **Episodic memory** | | | |
| *RL / RI* 16, learning (Z score) [37] | -0.89 (1.02) | -0.37 (0.84) | 0.52 |
| *RL / RI* 16, delayed recall (Z score) [37] | -0.97 (1.15) | -0.86 (1.31) | 0.98 |
| BVMT-R, learning (Z score) [38] | -0.36 (0.61) | -0.25 (1.36) | 0.34 |
| BVMT-R, delayed recall (Z score) [38] | -0.30 (1.38) | -0.32 (1.17) | 0.46 |
| **Attention / Working memory** | | | |
| D-KEFS—TMT 1 (Z score) [39] | -1.38 (1.17) | -0.92 (1.40) | 0.55 |
| D-KEFS—TMT 2 (Z score) [39] | -0.11 (0.91) | -0.78 (1.88) | 0.44 |
| D-KEFS—TMT 3 (Z score) [39] | -0.33 (1.31) | -0.05 (1.57) | 0.66 |
| Baddeley dual task paradigm (Z score) [40] | -1.78 (1.68) | -2.14 (1.87) | 0.52 |
| Spatial Span (Z score) [41] | - 0.17 (0.95) | -0.06 (0.8) | 0.83 |
| Digit Span, backwards (Z score) [33] | -0.06 (0.59) | 0.17 (1.29) | 0.74 |
| **Executive functions** | | | |
| D-KEFS—TMT 4 minus 5 (Z score) [39] | -0.83 (1.47) | -0.79 (1.54) | 0.37 |
| Six Elements task (Z score) [42] | -1.66 (0.36) | -2.07 (0.38) | 0.07 |
| **Global cognition** | | | |
| MoCA–Raw score [43] | 24.00 (2.19) | 23.83 (2.32) | 0.89 |
| MoCA–Z Score [43] | -1.24 (0.73) | -1.17 (0.53) | 0.83 |
| **Primary outcomes** | | | |
| **DEX (raw, self-rated)** | 21.17 (7.47) | 16.33 (5.05) | 0.21 |
| **DEX-C (raw, caregivers)** | 22.17 (10.88) | 14.8 (11.16) | 0.27 |

(*Continued*)

**Table 1.** (Continued)

|  | Group A Mean (SD) N = 6 | Group B Mean (SD) N = 6 | *p* |
|---|---|---|---|
| **ZMT total score (raw)** | 9.50 (2.74) | 9.17 (3.19) | 0.84 |
| **ZMT (raw errors)** | 3.60 (4.04) | 2.33 (2.34) | 0.53 |
| **Secondary outcomes** |  |  |  |
| **PDQ-39 total (%)** | 34.5 (10.93) | 25.67 (8.59) | 0.15 |
| **DRS-II total (raw)** | 137.00 (3.63) | 134.5 (6.03) | 0.41 |

*Notes.* Group A: GMT; Group B: psychoeducation; BJLO: *Benton Judgment of Line Orientations*; BVMT-R: *Brief Visual Memory Test*, *Revised*; DEX: Dysexecutive questionnaire; D-KEFS: Delis and Kaplan Executive Functioning System; DRS-II: Dementia Rating Scale, second edition; LED: *Levodopa Equivalent Dose*; MoCA: *Montreal Cognitive Assessment*; NAB: *Neuropsychological Assessment Battery*; PDQ-39: Parkinson Disease Questionnaire, 39 items; RL / RI 16: Rappel Libre / Rappel Indicé 16 items; TMT: *Trail Making Test*; WAIS-IV: Weschler Adult Intelligence Scale–IV; WMS-III: Weschler Memory Scale–III; ZMT: Zoo Map Test. Group equivalency was tested with T-tests (respective *p*'s reported in last column).

of a registered neuropsychologist and were conducted at each participant's home. All sessions were planned beforehand with each participant and occurred at an optimal moment of the day («ON» period, 60–90 minutes after participants took their medication). There were five different times of measure: BL took place just before the start of the intervention; Mid-point occurred after 3 weeks of intervention whereas FU occurred 1 week (FU-1), 4 weeks (FU-2), and 12 weeks (FU-3) after the intervention. The complete procedure is detailed in Fig 2. Independent raters (senior doctoral students), blind to the assignment of participants to groups, rated all protocols and entered data in the database.

In this study, primary efficacy outcomes are 1) subjective complaints of executive functioning, as measured by DEX (patient self-rated) and DEX-C (rated by caregivers); and 2) objective measures of executive functioning, as measured by Zoo Map Test (total raw score and number of errors). The DEX questionnaire and Zoo Map Test are both part of the Behavioral Assessment of Dysexecutive Syndrome battery (BADS) [44].

The DEX includes 20 items rated on a 5-point Likert scale from 0 (never) to 4 (very often). Higher scores suggest more problems with executive functioning. Both DEX versions were administered to patients (DEX) and caregivers (DEX-C) at BL (Baseline), mid-point and FU-1 (Follow Up-1), FU-2 and FU-3, respectively one, four- and twelve-weeks post-intervention. The DEX has good convergence validity, correlates with BADS scores and has previously been validated in PD patients [44].

*The Zoo Map Test (ZMT)* measures organization and planning capacities [43]. The ZMT scoring is based on the strategy used by the patient, on a 16-item scale (raw score) transformed into a profile score ranging from 1 (very poor strategy) to 4 (very good strategy). The number of errors was also recorded. ZMT was administered at BL and FU-1, FU-2 and FU-3. It was shown to have good construct validity and good correlation with other executive tests [44, 45].

Secondary outcomes include quality of life as measured by Parkinson Disease Questionnaire–39 items (PDQ-39) and global cognition as measured by Dementia Rating Scale 2nd edition (DRS-II).

*Parkinson Disease Questionnaire-39 items (PDQ-39)* is a self-rated questionnaire specifically designed for PD patients [46] to assess eight dimensions of quality of life: motor symptoms, activities of daily living (ADL), emotional well-being, stigmatization, emotional support, cognition, communication and body discomfort. Items are rated on a Likert scale from 0 (never) to 4 (always). Higher scores suggest lower quality of life. It was administered at BL, mid-point and FU-1, FU-2 and FU-3. PDQ-39 is widely used in research due to good construct validity, reliability and internal consistency [46].

*The Dementia Rating Scale- Second edition (DRS-II)* [47] is a brief neuropsychological instrument designed to assess general cognitive functioning and is validated to detect PD-MCI and PDD in French-speaking patients [48]. Because this study was conducted in Canada, the fluency task 'Names of USA states' (Initiation / Perseveration subscale) in the alternate version was changed for 'Boys Names'. A cut-off score of ≤ 140 to detect PD-MCI was shown to have good sensitivity and specificity values (respectively, 0.85 and 0.54) with this translated version of the DRS-II [48]. It includes five subscales: Attention, Initiation/Perseveration, Construction, Concepts and Memory. Raw scores range from 0 to 144 (respective cut-offs for PDD ≤ 132 and PD-MCI ≤ 140) [48]. The original version was administered at BL and FU-2, and the alternate version was administered at FU-1 and FU-3 to avoid practice effect.

## Statistical analyses

Descriptive statistics (mean and standard deviation) were calculated for socio-demographic characteristics of participants, all screening scores and BL variables. Then, T-tests for independent samples were conducted to test for statistical differences between groups. Differences between BL data and all other repeated measures were calculated for primary (DEX, DEX-C and ZMT) and secondary outcomes (DRS-II, PDQ-39). All outcomes were analyzed using a repeated measure ANCOVA (linear mixed model), in which the terms Group (Group A or B), Time (1: BL; 2: mid-point; 3: FU-1; 4: FU-2 and 5: FU-3), the interaction Group*Time, and the covariate (disease duration) were considered as fixed effects. The best correlation matrix between participants' observations taken over time was selected using the Akaike Information Criterion. In all cases, it was a first order autoregressive structure, AR(1), for unequally spaced time points that best fit the data. Effect sizes ($\eta^2$) were calculated. Interpretation of effect sizes follows Cohen's [48]: 0.00 = no effect; 0.01 = small effect; 0.06 = intermediate effect. Following a significant effect of any sources of variation, post hoc comparisons (protected LSD) were conducted to determine which treatment combination differs. The normality assumption was verified using the Shapiro-Wilk's statistic and the homogeneity of variances was investigated visually using the residuals plot. All statistical analyses were performed with SAS software (version 9.4 for Windows) at the α = 0.05 level of significance and were verified by a professional statistician.

## Results

Clinical and sociodemographic characteristics of the final sample, as well as screening neuropsychological evaluation scores and BL scores of each group, are shown in Table 1. At BL, groups show some differences. It seems that Group A comprises older and more educated participants, with scores slightly more elevated regarding Hoehn and Yahr (H&Y) stages, Levodopa equivalent dosage (LED) and PDQ-39 scores. The performance on lexical and category fluency tests seems inferior in Group A compared to Group B. The non-significant p-values for these results are likely due to a lack of power to detect differences (given the small sample size). The only statistical difference is that Group A suffers from the disease since a significant longer period than Group B. Disease duration was therefore considered as a covariate in statistical analysis. Otherwise, most participants were retired: four in Group A and four in Group B. Other participants were working part-time or full-time. Missing data were not replaced, and all available data were included in analyses.

### Feasibility

**Acceptability.**   Overall, it seems that both interventions were considered acceptable by all participants and most caregivers. The only caregiver who was unable to participate in the

study had a severe impairment (major chronic illness). Three participants out of 12 (one in Group A and two in Group B) mentioned they would have liked to continue the interventions. Most participants said they liked the intervention and liked the fact that the trainer came at home. At FU-3, two participants in Group A and three in Group B said they were now frequently using some of the exercises learned during the sessions (ex. Mindfulness, STOP! For Group A, etc.). All participants found the intervention appropriate for their condition.

**Demand.** All participants and screened patients expressed their interest in the intervention. Most of them (n = 11) considered they wanted to develop new tools to better cope with PD-MCI. Caregivers expressed the need to understand cognitive impairment.

**Implementation.** During the intervention sessions, all participants were able to execute the exercises and reflections appropriately. However, a participant in Group B fell asleep during a mindfulness exercise. Participants of Group A were given approximately 60 minutes of exercises to practice between intervention sessions each week. All but two participants in Group A did the between-session exercises as suggested by the trainer. The drop-out participant did only 30 minutes of practice per week while another participant did more practice than the suggested amount of time (150 minutes per week). Participants of Group B have not been asked to do exercises between sessions, but three participants did the mindfulness exercises between sessions and requested supplemental books or resources.

**Practicality.** The clinician/trainer needed material to carry out the interventions and evaluations: a computer and an iPad to conduct the interventions and several neuropsychological tests (listed in results) to assess the participants. As mentioned earlier, each participant was given a workbook. Groups A and B had different workbooks. The trainer also needed a mean of transportation. Most participants were living in a 20-km area.

Through the entire study, the trainer worked eight hours for the neuropsychological assessment and six hours in intervention sessions for each participant. In addition, the trainer travels a total of four hours per participant (transportation expenses). Each participant thus required 20 hours of work from the clinician. In Canada, a junior neuropsychologist in a private setting typically charges 130$ CAD per hour for service at home. For one participant: 20 * 130 = 2600 $ CAD which translates into 2151,69$ USD.

## Safety

One participant in Group A was unable complete the study after mid-point because of the sudden death of a family member. A Group B caregiver was unable to complete any part of the study due to illness. No change in LED occurred in any patients during intervention. However, LED changes occurred in three participants at follow-up: two Group A participants reported a dosage increase (24.5% of LED at FU-2 and 16.6% of LED at FU-3, respectively), whereas a Group B participant reported the introduction of amantadine (12.6% total increase in LED at FU3). There were four missed sessions in Group A and two in Group B. All participants informed the trainer before the session that they were having last minute difficulty and asked her to postpone the missed session. All reported sessions were retaken.

Before each session, Group A participants presented a mean fatigue score of 4.25 /10 (SD = 1.40) which indicates mild-to-moderate fatigue. After each session, they registered a mean fatigue score of 5.29/10 (SD = 1.35) which indicates moderate fatigue. Before each session, Group B participants had a mean fatigue score of 3.34/10 (SD = 1.77) which indicates mild fatigue. After each session, they reported a mean score of 3.97/10 (SD = 1.9; mild fatigue). Overall, participants of Group A expressed slightly more fatigue before and after sessions than participants of Group B. However, the fatigue remained mild to moderate throughout the study as no patient reported severe fatigue.

**Table 2. Results of the repeated measure ANCOVAs for primary and secondary outcomes.**

| Fixed effect (*df1, df2*) | DEX-p F-values (p-values) *Effect size ($\eta^2$)* | DEX-C F-values (p-values) *Effect size ($\eta^2$)* | ZMT (raw) F-values (p-values) *Effect size ($\eta^2$)* | ZMT (errors) F-values (p-values) *Effect size ($\eta^2$)* | PDQ-39 F-values (p-values) *Effect size ($\eta^2$)* | DRS-II F-values (p-values) *Effect size ($\eta^2$)* |
|---|---|---|---|---|---|---|
| Group (1, 10) | 0.82 (0.388) ***0.008*** | 0.36 (0.564) ***0.006*** | 0.21 (0.653) ***0.006*** | 1.19 (0.316) ***0.082*** | 3.08 (0.109) ***0.073*** | 1.42 (0.261) ***0.04*** |
| Time (4, 36) | 2.96 (0.033) * ***0.087*** | 6.02 (0.017) * ***0.047*** | 0.92 (0.443) ***0.07*** | 16.66 (0.001) * ***0.102*** | 1.77 (0.156) ***0.017*** | 0.23 (0.877) ***0.003*** |
| Time*Group (4, 36) | 0.75 (0.567) ***0.008*** | 0.483 (0.749) ***0.001*** | 0.56 (0.646) ***0.026*** | 3.53 (0.064) ***0.057*** | 5.31 (0.002) * ***0.066*** | 0.44 (0.723) ***0.009*** |
| Disease duration (1, 36) | 1.60 (0.214) ***0.007*** | 17.27 (0.004) * ***0.0005*** | 1.76 (0.196) ***0.003*** | 6.19 (0.03) * ***0.066*** | 0.12 (0.734) ***0.004*** | 5.48 (0.027) * ***0.05*** |
| MCID (SEM) | 3.054 | 4.552 | 1.116 | 1.315 | 4.463 | 2.483 |

*Notes.*

* $p < 0.05$.

Interpretation of effect sizes follows Cohen's [55]: 0.00–0.003 = no effect; 0.01–0.039 = small effect; 0.06–0.11 = medium effect; 0.11–0.2 = large effect. *Notes*. DEX-p: Dysexecutive questionnaire for patients; DEX-C: Dysexecutive questionnaire for caregivers; df: degrees of freedom (1: numerator; 2: denominator); DRS-II: Dementia Rating Scale, second edition; MCID: Minimal Clinical Important Difference (for an indication of change); PDQ-39: Parkinson Disease Questionnaire, 39 items; SEM: Standard Error Mean; ZMT: Zoo Map Test.

## Preliminary efficacy testing on primary outcomes: Repeated measures ANCOVAs

Table 2 presents all results for primary and secondary outcomes.

**DEX self-report.** Results show a significant difference on the DEX between measurement times, for both groups ($F(4,36) = 2.96$, $p = 0.033$) [CI 95%: 10.75–15.23]. The effect size is considered medium ($\eta^2 = 0.087$). However, the results are not significantly different between the two groups, for all times of measurement ($F(1,10) = 0.82$, $p = 0.388$;). The interaction effect (Group*Time) is not significant ($F(4,36) = 0.75$, $p = 0.567$). These results mean that the improvement occurring overtime on the DEX are not significantly different between groups. Furthermore, disease duration, as covariable, has no significant effect on the outcomes between groups ($F(1,36) = 1.60$, $p = 0.214$). Post-hoc analysis shows that the changes between time periods occurred between BL and FU1; and BL and FU2. Taken together, these results show a decrease in both groups on DEX scores following the interventions, which means that patients report significantly better executive functions after receiving either of the two treatments.

**DEX-C.** A first analysis shows there is no evidence that DEX-C scores change after either intervention, even when adjusting for disease duration ($p > 0.05$ for all fixed effects). However, an outlier was identified (Z score > 3 at baseline and all other measurement times). After removing this outlier from the DEX-C data, analyses were reconducted. Without the outlier, results show a significant difference between measurement times on DEX-C scores for both groups ($F(4,36) = 6.02$, $p = 0.017$) [CI 95%: 9.63–17.23]. The effect size is considered small ($\eta^2 = 0.047$). However, groups are not significantly different between the times of measurement ($F(1,10) = 0.36$, $p = 0.564$). The interaction effect (Group*Time) is not significant either ($F(4,36) = 0.483$, $p = 0.749$), which means that the improvement in DEX-C scores are not significantly different between groups. Disease duration has a significant effect between groups (F(1,36) = 17.27, $p = 0.004$) which means that patients with the longest disease duration, when rated by caregivers, improve more executive functions than those with shortest disease duration.

**ZMT.** ZMT total raw scores do not significantly change after either intervention, even when adjusting for disease duration ($p > 0.05$ for all fixed effects). However, patients of both

groups significantly make fewer ZMT errors between measurement times ($F$(3,36) = 16.66, $p$ = 0.001) [CI 95%: 1.07–2.93]. The effect size is considered medium ($\eta^2$ = 0.102). Groups are not significantly different at the different times of measurement ($F$(1,10) = 1.19, $p$ = 0.316). The interaction effect (Group*Time) is not significant ($F$(4,36) = 3.553, $p$ = 0.06). Thus, both interventions significantly help to reduce the number of errors in ZMT, although there is a tendency for the GMT group to have more linear results than the other group, in this regard. It is interesting that 5/5 (100%) of Group A (GMT) participants improve or maintain their profile scores at FU-1, FU-2 and FU3. On the other hand, 3/6 (50%) of Group B participants improve or maintain their profile scores at FU-1, whereas 4/6 (66.7%) participants improve or maintain their profile score at FU-2 and FU-3.

### Preliminary efficacy testing on secondary efficacy outcomes: Repeated measures ANCOVAS

**PDQ-39.** There is a significant interaction between measurement times and group condition on PDQ-39 ($F$(4,36) = 5.31, $p$ = 0.002) [CI 95%: 15.33–25.61]. The effect size is considered medium ($\eta^2$ = 0.066). Disease duration has no significant effect between groups ($F$(1,36) = 1.60, $p$ = 0.214). Taken together, these results show that PDQ-39 scores improve in Group B, whereas PDQ-39 scores are maintained in Group A (Fig 3E). Post-hoc analysis shows that changes between groups occurred between BL and mid-point. After mid-point, scores were maintained in both groups.

**DRS-II.** There is no change in DRS-II scores after either intervention, even when adjusting for disease duration ($p$ >0.05 for Measurement Time, group and interaction Measurement Time*Group). However, disease duration seems to have an effect between groups ($F$(1,36) = 5.48, $p$ = 0.027), with patients with longest disease duration performing worse on DRS-II than patients with shorter disease duration. The effect size is considered medium ($\eta^2$ = 0.05).

## Discussion

The objective of the present study was to compare feasibility, safety and preliminary efficacy of GMT and PSYCH-Mind to treat executive dysfunction in PD-MCI.

### Safety

There was no adverse event related to the administration of interventions. Changes in medication were reported for three participants, two in GMT group and one in PSYCH-Mind group, and that perhaps could explain some results. These changes were reported at FU-2 or FU-3 and ranged from 12 to 24% increases in LED. However, a review on the effects on cognition of anti-Parkinson medications in PD-MCI has found that such changes on short periods did not significantly affect patients' cognitive functioning [5]. Since positive outcomes were mostly reported on DEX at FU-1, changes in LED can hardly explain the positive results. Fatigue was tolerable, even though Group A overall reported a little more fatigue than Group B. However, the baseline level of fatigue was higher in Group A, which is expected considering that their disease was slightly more advanced than in Group B.

### Feasibility

Concerning acceptability, it seems that both interventions were acceptable and appreciated by patients, caregivers, and the clinician trainer. As for the demand, participants did use in their daily life the strategies and information learned and discussed during the intervention sessions. Both interventions and trainer-recommended exercises were well executed and implemented.

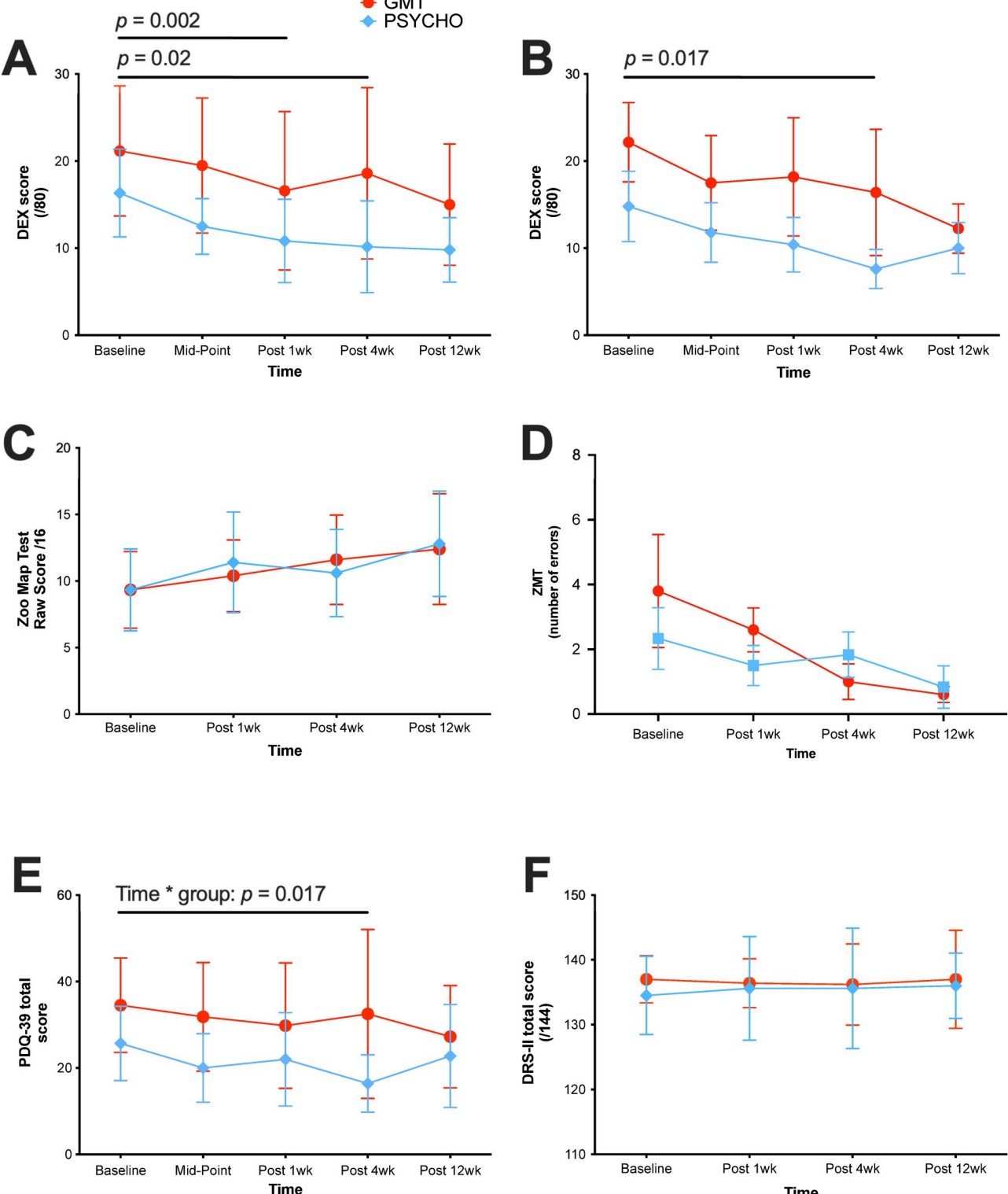

**Fig 3. Effects of cognitive interventions on primary and secondary outcomes.** (A) *DEX (self-rated)*, (B) *DEX (caregiver, all data included)*, (C) *ZMT raw scores*, (D) *ZMT number of errors*, (E) *PDQ-39 and* (F) *DRS-II scores overtime, presented as mean +/- SD.* Group A (GMT) n = 5; Group B (PSYCH-Mind) n = 6.

Regarding practicability, cost versus benefit estimates of the Adapted-GMT and PSYCH-Mind seems reasonable when compared to those of the classical drill-and-practice cognitive training programs. In the present study, one trainer utilized one computer and an iPad in addition to a neuropsychological battery including tests and questionnaires. For 12 participants, most of the costs were thus due to the salary and transportation expenses of the clinician/trainer. In the typical drill-and-practice cognitive training conducted individually or in group, each participant requires a computer, and must learn how to use the computer to properly run the training program. Furthermore, it often requires the assistance of a coach or trainer (implying a salary to pay) to help participants start the program, to monitor assiduity and difficulties of participants. For the cognitive training practiced in group, one must also consider the cost and difficulty of transportation to the meeting place for each participant. Assiduity may possibly be better when interventions are conducted individually at each patient's home, because it reduces the difficulty and fatigue of transportation often experienced by patients with PD. The results of the present study versus those of previous studies at least partly support this affirmation. Indeed, previous studies reported that 88% of participants completed 75% of the intervention [49], whereas in the present study, all but one participant (11/12 = 91,6%) completed 100% of the sessions. A possible explanation for better assiduity is the duration of each session and of the program. In previous studies, each session was lasting from 30 minutes to 120 minutes and administered over periods from 8 to 24 weeks whereas in the present study, each session lasts from 60 to 90 minutes and the two programs were administered in five weeks. Overall, these results show that both interventions were feasible.

Moreover, the results obtained by PD-MCI after the Adapted-GMT do compare with results obtained previously by participants with other neurological conditions who completed the original GMT. Indeed, in the Stamenova and Levine meta-analysis [17], the effect sizes of improvement after GMT as measured by DEX or Cognitive Failure Questionnaires were respectively small to medium for patients and medium for the caregivers. In the present study, the effect sizes of improvement on both DEX and ZMT-errors were respectively medium for patients and small for the caregivers (DEX-C).

Furthermore, the current results are at least comparable if not better than those obtained in previous studies on cognitive training in PD-MCI. Indeed, the effect sizes registered here are small to medium whereas in the previous PD studies, they are null or rather small [9, 10]. A possible explanation for this better effect size is the strategy-based (top-down) approach used in the present study as opposed to the drill-and-practice (bottom-up) approach used in prior research. This will be further discussed in the next section.

## Preliminary efficacy testing

The results on the two DEX versions indicate that both intervention programs have a positive impact overtime on executive complaints by patients and caregivers. This finding is supported by the significant decrease in errors made on the ZMT by participants of Groups A and B, ZMT being an objective measure of organization and planning functions. However, there was no significant improvement in ZMT total raw scores in both groups, although 100% participants who had received Adapted-GMT improved or maintained their profile score (not reaching statistical significance), even at FU2 and FU3, whereas 67% participants of the PSYCH-Mind group, maintained or improved their profile score after 4 and 12 weeks. Both interventions included education on cognitive symptoms and mindfulness exercises. However, only Adapted-GMT comprises reflective exercises and goal fulfilling strategies during and between the intervention sessions. This content primarily distinguishes the Adapted-GMT from the PSYCH-Mind program and could perhaps explain the slightly different improvement

profile between the two groups on the ZMT. Altogether, the DEX and ZMT results suggest that both patient groups may have benefited in some way from the interventions. Interestingly, patients with longer disease duration experienced improvement while their DRS-II scores were inferior to those with shorter disease duration, suggesting they benefited from interventions despite greater overall cognitive impairment.

Only Group B improved quality of life (PDQ-39) four weeks after intervention. This could be explained, at least partly, by the format and material of PSYCH-Mind sessions. They were indeed designed as discussions in which there was information sharing with patients and caregivers about a wide range of different PD symptoms, and there were more opportunities to ask questions than in Group A. By contrast, Adapted-GMT focused exclusively on giving information and teaching strategies to remediate executive deficits. It is noteworthy that although the PSYCH-Mind participants were not asked to practice mindfulness exercises between sessions, half of them reported doing so. This is a positive indicator of both acceptability and demand. Neither intervention influenced overall cognitive functioning of the patients (DRS-II).

To our knowledge, this is the first study conducted in PD-MCI patients to investigate the effects of GMT which taps into top-down processes. Previous cognitive training programs specifically aimed at improving executive functions in PD-MCI involved computerized or paper and pencil exercises, mostly 'drill and practice' type of training triggering bottom-up processes [9, 10, 49]. Two very recent studies administered traditional drill-and-practice (bottom-up) cognitive training with a particular focus on improving working memory [50], logical executive functions, episodic memory, attention and processing speed [51]. Ophey et al. [50] assessed the effects of a 5-week individualized computerized training targeting working memory on 75 cognitively healthy PD patients. The participants were 64 years old and had 15 years of education. They were thus younger than the present sample but had approximately the same level of education. Three months after the end of the training, there was only a small effect size on the test of verbal working memory [50]. Bernini et al. [51] assessed the effects of a 3-week computerized cognitive training targeting logical executive functions, episodic memory, attention and processing speed. The 57 participants were 74 years old with 8 years of education, which is in the same age group but with an inferior level of education than the present sample. After the end of the intervention, participants recorded improvements in global cognition, attention and processing speed as measured by neuropsychological tests [50]. Importantly, the improvements were not maintained after six months. These recent studies show positive but modest results as measured by 'laboratory' neuropsychological tests, as opposed to the self-measure of executive functioning, evaluating daily functioning, used in the present study [50–52].

Thus, very few of these studies intended to learn strategies to PD patients triggering top-down processes [10, 53–55]. As a result, the previous cognitive training studies generally showed limited improvements on trained tasks, with very little if any transfers in daily activities [9, 10]. The need to transfer the benefit of cognitive training to daily life tasks is supported by the study of Vlagsma et al. [55] who showed that PD patients expressed the same rehabilitation goals, anchored in routine daily tasks than dysexecutive patients with acquired brain injury. Before the present study, only two trials including PD patients utilized cognitive rehabilitation to address the issue of improving cognition in daily functioning. Cognitive rehabilitation is a type of intervention for patients with dementia that focus on individualized goals to improve one or more specific activities of daily activities (ADLs). To do this, the clinical instructor trains the patient in the use of restorative and compensatory cognitive techniques, has him apply these techniques to one or more ADLs to be improved, and has him practice several times a day and a week. As a result, the specific ADL that has been targeted may improve, but the improvement usually does not generalize to other ADLs. In the first study

[56], 240 PD patients with any MCI profile were randomly allocated to one of the three training groups: Group A received cognitive training (paper and pencil, and computer) only; Group B received the same cognitive training in addition to transfer (rehabilitation) training whereas Group C also received the same cognitive training in addition to transfer and psychomotor trainings. After six months of training, Group C registered the best improvement on two measures of overall cognitive functioning, the ADAS-Cog and the SCOPA-Cog, as well as on quality of life. More recently, Hindle et al. [54] administered to patients with PDD a goal-oriented cognitive rehabilitation program. The authors reported that after two months of treatment, results of cognitive rehabilitation were better than results of treatment-as-usual and relaxation therapy for the primary outcomes of self-rated goal attainment. After six months, the outcomes of cognitive rehabilitation remained superior to those of treatment-as-usual. These last two trials positively impacted the daily functioning of PD patients. However, the first one was very costly with an implementation in both a rehabilitation centre and at home, and requiring, among other materials, a computer for each patient in both locations. In addition, it involved a physiotherapist for the patient, a coach for the caregivers [55]. The second study administered a less costly program, but that was designed for patients suffering from dementia, which is not appropriate for PD-MCI [54].

Therefore, cognitive training programs for PD-MCI including strategy learning to transfer in various situations and tasks of daily life are still scarce but available data show promising results. The present study is the only one, though, which evaluated the effect of cognitive interventions using both inputs of participants and caregivers, the latter being considered a generalization measure [48]. Indeed, these changes on the DEX can be considered as an effectiveness clinical outcome, given that both patients and caregivers reported significant changes in their daily lives.

Only one previous study conducted in PD patients combined psychoeducation with mindfulness in intervention sessions. This study showed modest but, like our trial, positive results on quality of life. In addition, the authors reported an improvement of depressive symptoms [27]. This was not the case in our study, probably because PD patients were not included if, besides PD-MCI, they met the diagnostic criteria for a psychiatric disorder like major depression. The program of Advocat et al. [27], as well as the standardized psychoeducation program for PD, the Patient Education Program Parkinson (PEPP), do not include education on cognitive symptoms [21]. Studies on the PEPP show that it increases mood and reduces burden among PD patients and caregivers [21–23], but it does not provide information on the impact of the PEPP on cognitive functions, contrary to the present PSYCH-Mind program. As for mindfulness administered alone, none of the previous studies included PD-MCI participants, nor did they give information on their cognitive decline [18]. It is noteworthy that sessions combining psychoeducation with mindfulness exercises resulted in a decrease of executive complaints, as well as an increase in quality of life in the present study. This seems to be the first study investigating the combined effect of mindfulness and psychoeducation on cognitive symptoms in PD-MCI.

What could be the possible neurological mechanisms underlying the effects of these top-down interventions? Recent literature might give a possible explanation. A recent study explored the neurological effects of a working memory cognitive training in a patient [57], with PET scan technology. After the training, they showed a significant decrease in the activation of the frontal-parietal circuit during tests requiring working memory, showing that these tests required less and less resources for the patient [57]. This is one of the first studies to show a link between the activity of a neural circuit and a form of cognitive training in PD. In short, it seems that the strategy-learning could have an impact on neural networks involved in cognitive deficits.

According to literature PD-MCI with executive dysfunctions is often linked to dopaminergic decrease in the mesocorticolimbic pathways [1–4]. In the present study, it is possible to suppose that these pathways might be activated more efficiently with top-down interventions.

Present results show the importance of considering participant's clinical profile and demographics to personalize treatment. Previous studies included participants aged between 58.8 and 70.9 years old, which is slightly younger than the present sample, with shorter disease duration (between 2 and 8 years), and similar disease stage (H&Y 2–3). However, they provided more training sessions per week (2 to 4, each lasting 30–90 minutes) as well as more sessions for the entire programs (10–16 sessions over 5–12 weeks) and obtained similar or less benefit compared to the present study [9, 10]. Older PD participants whose illness lasts longer may be more easily tired and need more time to recover between training sessions than younger counterparts. Thus, the present participants may have benefited from a reduced number of sessions (5 sessions) delivered over the same amount of time (5 weeks) than previous studies to improve after Adapted-GMT and PSYCH-Mind programs. The spaced training schedule associated with home administration possibly helped to counter the effect of age and fatigue, even if the duration of each session was comparable to that in previous studies.

A last important aspect to consider is the opinion of caregivers included, especially those who share the patient's everyday routine. The decrease in DEX-C scores registered in both groups was maintained over a three-month period, except for that of a caregiver who was excluded from analysis. This particular Adapted-GMT caregiver observed more important motor deterioration in her husband' symptoms. As a result, the patient medication had to be adjusted between FU1 and FU2, and there was a significant increase of PDQ-39 and DEX and DEX-C scores, indicating a deterioration of executive functions and quality of life. In other words, the disease condition of this patient worsened over the course of the study. Previous studies on cognitive training in PD participants often do not include questionnaires answered by caregivers about the patient's cognitive abilities [9, 58]. However, the perception, by the caregiver, of the progress made by the patient, can eventually have an impact on his own feeling of burden.

Although potential lack of statistical power is a major caveat, the present trial does present some strong points. First, interventions were tailored to the needs of participants with executive dysfunctions, especially more important in Group A. The exhaustive neuropsychological evaluation at screening ensured that the intervention matched the participant's cognitive profile.

Secondly, this is among the first studies to assess effects of two cognitive interventions at home that provided the same time and attention to all participants; one program directly targeted improving executive functions while the other attempted to improve cognition through education and the practice of mindfulness. Results show that both programs with a top-down approach could be effective in improving day-to-day executive functioning in PD-MCI patients. Furthermore, participants from both groups mentioned enjoying the interventions indicating that patient-tailored interventions at home are not only feasible, well tolerated, and safe but can also be enjoyable. Both interventions were easily implemented and administered by a clinician.

Another advantage is that both interventions can be considered more economical than bottom-up computerized cognitive training administered in groups. Indeed, given the price of the material and software needed for computerized training, as well as the difficulties and transportation expenses for the patients, Adapted-GMT and PSYCH-Mind are more economical for the potential benefits of the interventions. Given the small drop-out rates and high adherence, these interventions seem also more advantageous. Future research shall continue to examine which participants benefit the most to a given intervention.

Finally, home sessions seem better for therapeutic alliance, anxiety and treatment compliance [12]. In fact, many participants mentioned they appreciated the therapist coming to their home.

## Limitations

The major limitation of this study is obviously the small number of participants, which does not allow drawing general conclusions applicable to all PD-MCI patients. Lack of statistical power might also explain why there was no significant difference between the two groups in ZMT errors and raw scores after the intervention. However, statistical power was sufficient for the principal objectives of this study, which were to assess feasibility and safety as well as preliminary efficacy.

Because of the small sample size, it is not possible to assume group equivalency. The non-significant *p*-values regarding patients' characteristics are likely due to a lack of power to detect differences. Indeed, the groups show a few differences that fit with the significant longer disease duration presented by participants of Group A. This finding suggests that progression of the disease is more advanced in Group A. For instance, the fatigue reported in Group A might be explained by the more advanced disease stage. However, it is interesting to underline that the effect of Adapted-GMT is significant in this group on both DEX scores and ZMT errors, despite the fact that participants suffer from the disease since the longest time.

The short duration of intervention, despite being an advantage for vulnerable and frail patients, might also be considered as another possible limitation. A five-week program might not be long enough to have a significant, lasting impact on neuropsychological tests [49], but it is quite typical of feasibility studies [31]. In previous studies assessing the effects of cognitive intervention in PD, most programs lasted between 4 and 12 weeks [9, 10] and often were administered to PD patients without MCI [9, 10]. However, it was important in the present study to offer the shortest possible program, so PD progression or medication changes could be better (at least partly) controlled. On the other hand, GMT studies tend to calculate the number of training hours more than the number of sessions [17]. In the present trial, Adapted-GMT group received from 12 to 20 hours of total training hours (including between sessions exercises), which is average compared to other GMT studies giving approximately a mean of 15 hours of training [17]. Therefore, training time in the Adapted-GMT group seems adequate in the present study.

Another, but important limitation concerns the blinding procedure which could cause an investigator bias. The same clinician was indeed in charge of all interventions and was aware of the allocation of participants in each group. However, that procedure was chosen because it allows to control for the trainer characteristics, such as therapeutic alliance [59].

At last, a performance bias could have affected the way participants acted through the study. For instance, this bias may have prompted them to exercise more often between sessions to please the research team.

## Conclusion

Although cognitive symptoms in PD are frequent and have adverse consequences for patients and caregivers, available data on cognitive interventions are still scarce, and accessibility to tailored cognitive treatments is restrained. Both interventions were acceptable, feasible, easily implemented and showed satisfactory preliminary efficacy. According to the present results, interventions could be suggested to patients aged between 50 and 75 years old, with mild to moderate motor symptoms and moderate executive complaints. It seems also that patients with a longer disease duration can complete the Adapted-GMT sessions without too much

fatigue. In the future, the effects of these two top-down programs should be assessed in larger samples of PD-MCI with executive dysfunctions and presenting with other clinical characteristics.

## Supporting information

**S1 Checklist. CONSORT 2010 checklist of information to include when reporting a randomised trial**[*].
(DOCX)

**S1 File.**
(DOCX)

**S2 File.**
(ZIP)

## Acknowledgments

The authors wish to thank Dr Brian Levine and Baycrest Centre for granting permission to use the GMT material, and Dr Leslie Fellows and her team at McGill University for their contribution to the GMT translation in French. Finally, the authors wish to thank *Réseau Parkinson Québec* as well as patients and families who accepted to participate in this study.

## Author Contributions

**Conceptualization:** Ariane Giguère-Rancourt, Martine Simard.

**Data curation:** Ariane Giguère-Rancourt, Marika Plourde, Eva Racine, Marianne Couture.

**Formal analysis:** Ariane Giguère-Rancourt.

**Investigation:** Ariane Giguère-Rancourt, Marika Plourde, Eva Racine, Marianne Couture, Nicolas Dupré, Martine Simard.

**Methodology:** Ariane Giguère-Rancourt, Martine Simard.

**Project administration:** Mélanie Langlois, Nicolas Dupré, Martine Simard.

**Resources:** Martine Simard.

**Software:** Martine Simard.

**Supervision:** Mélanie Langlois, Nicolas Dupré, Martine Simard.

**Validation:** Martine Simard.

**Writing – original draft:** Ariane Giguère-Rancourt.

**Writing – review & editing:** Ariane Giguère-Rancourt, Marika Plourde, Eva Racine, Marianne Couture.

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
