## [Decision Letter · Decision Letter 0]

5 May 2021

PONE-D-21-08484

Goal Management Training and Psychoeducation for Treatment of Executive Dysfunction in Parkinson’s disease: A Feasibility Pilot Trial.

PLOS ONE

Dear Dr. Giguère-Rancourt,

Thank you for submitting your manuscript to PLOS ONE. After careful consideration, we feel that it has merit but does not fully meet PLOS ONE’s publication criteria as it currently stands. Therefore, we invite you to submit a revised version of the manuscript that addresses the points raised during the review process.

We look forward to receiving your revised manuscript.

Kind regards,

Walid Kamal Abdelbasset, Ph.D.

Academic Editor

PLOS ONE

Journal Requirements:

2. Thank you for submitting your clinical trial to PLOS ONE and for providing the name of the registry and the registration number. The information in the registry entry suggests that your trial was registered after patient recruitment began. PLOS ONE strongly encourages authors to register all trials before recruiting the first participant in a study.

1) your reasons for your delay in registering this study (after enrolment of participants started);

2) confirmation that all related trials are registered by stating: “The authors confirm that all ongoing and related trials for this drug/intervention are registered”.

4. Please include a copy of Table 1 which you refer to in your text on page 7.

Reviewers' comments:

Reviewer's Responses to Questions

**Comments to the Author**

1. Is the manuscript technically sound, and do the data support the conclusions?

Reviewer #1: Yes

Reviewer #2: Partly

Reviewer #3: Yes

2. Has the statistical analysis been performed appropriately and rigorously? 

Reviewer #1: Yes

Reviewer #2: Yes

Reviewer #3: Yes

3. Have the authors made all data underlying the findings in their manuscript fully available?

Reviewer #1: Yes

Reviewer #2: Yes

Reviewer #3: Yes

4. Is the manuscript presented in an intelligible fashion and written in standard English?

Reviewer #1: Yes

Reviewer #2: Yes

Reviewer #3: Yes

5. Review Comments to the Author

Reviewer #1: Thank you for giving me the opportunity to review this article.

Abstract:

1. Mention the acronym of abbreviations when it is used for first time. (GMT, PDQ, DEX, ZMT etc..)

2. Include the study design, randomization and allocation procedure.

3. Mention the treatment for the both groups.

4. Results part should be more informative including CI 95% with p scores.

5. Make the conclusion more precise and avoid abbreviations.

Introduction

6. Please change the reference pattern as per author guidelines.

7. Please avoid subtitles.

8. How come this pilot study is differing from (McLean et al., 2017 and Stamenova & Levine, 2019 studies?

9. The introduction part fails to show the research gap and prove the novelty with recent references.

10. Add the clinical significance of this article over the participants and researchers.

Methods

11. Include the clinical trial registration number.

12. Include the study design, randomization and allocation procedure in detail.

13. Mention the selection criteria in detail (exclusion).

14. Include the intervention procedures in detail for study repetition.

15. Whether the interventions are performed at clinic? If performed at clinic means, how they are monitored?

16. Include the reliability and validity of outcome measures with references.

17. What is the need of doing sample size calculation as it is a feasibility pilot trial.

18. Mention about the blinding procedure.

Results

19. Include the information about the test used for analyzing homogeneity and its interpretation.

20. Include the reports with CI 95% with p scores.

21. Mention the effect size of primary and secondary variables and its MCID scores.

Discussion

22. Summarize the discussion part.

23. The discussion part should discuss the relation between the outcome variables and PD patients with latest references.

24. Make the conclusion more precise and avoid abbreviations.

25. Future recommendations and clinical significance of the study is missing.

Reviewer #2: In this study, the authors examined feasibility and preliminary efficacy of two home-based cognitive interventions in patients with Parkinson’s disease and mild cognitive impairment. Six patients experiencing executive dysfunction were randomized to each intervention, which consisted of weekly 60-90 minute sessions for five weeks. Measures were collected at baseline, mid-point, one-week, four week and 12-week follow-ups. Executive functions were assessed by the DEX questionnaire and Zoo Map Test. Secondary outcomes were quality of life and presence of psychiatric symptoms.

On the Consort checklist, the authors have entered that information on the registration of the trial, and the availability of the trial protocol, are “N/A.” This information (as follows) is provided in a letter to the editor, but should also be included in the manuscript:

“In the compressed file uploaded, you will find the originial protocol that was accepted by the Ethical Committee of CHU de Québec - Université Laval, as well as the last confirmation letter sent by the ethical committee. As you will see, the original documents are in French. In order to provide an english translation, we attached the Clinicaltrial.gov registration that explains the ethically approved protocol in English. The clinicaltrial.gov number of the trial is: NCT04636541.”

The study has a number of strengths. The authors have done an excellent job of justifying and describing the tests used in the study, and of explaining the test modifications and reasons for these in the cohort. Obtaining neuropsychological measurements at baseline is also a strength of the study. The analyses are clearly described and the paper has been nicely and clearly written. I do have some comments and questions, however.

Methods

• Were the same mindfulness exercises used for both groups?

• Did authors adjust for time spent doing mindfulness exercises between sessions in the GMT group?

• In the psychoeducation group? Did these differ between the two groups?

• Are there different versions of the ZMT or are patients repeating the same test/questions each time they are evaluated?

• Power analysis: What “effect size” in the study did the authors base the power calculation on? Change in DEX scores? Change in ZMT? Feasibility (if so, please define how this was measured in terms of sample size calculation)? Was the study powered for feasibility or efficacy? The authors should clarify. Also, please explain what a “medium” effect size means in terms of the quantitative measurement scores/changes (or whatever the primary measure was on which the power was calculated).

Results

• The authors should not assume that the two groups are comparable at baseline. In terms of LED and some of the Screening Neuropsychological Evaluation tests (Lexical fluency, NAB, e.g.), results appear quite different, and the non-significant p-values are likely due to a lack of power to detect differences (given the very small sample size). This should be acknowledged in the results in addition to being discussed as a limitation.

• The authors state in the Methods that non-parametric tests were used to compare groups. Data in Table 1 should therefore be summarized with non-parametric measures (median, IQR). If variables are actually normally distributed, they should be summarized with means and standard deviations and compared using parametric tests, which are more powerful when data are normally distributed.

Results

• In presenting the results, are the authors equating safety with feasibility? Given that “the aim of this pilot study was to test ease of implementation, differential safety and preliminary efficacy,” a section that addresses feasibility should be presented in the results section, probably prior to the sections on efficacy (for which the study was not really powered, I don’t think?).

• Results for DEX (self-report) are given “regardless of group.” Do the authors mean that all twelve patients were analyzed together? This should be more clearly stated, and analyses over the entire group should be distinguished from analyses between groups throughout the Results section.

• Post-hoc analyses should not be done if the omnibus test is not significant. I think it is fine for the authors to give these data descriptively, without p-values attached/interpreted.

• Again, given the very small sample size, an absence of difference between groups has not been shown, as this could be a power issue. The study has not been powered to detect interactions, which would be difficult to see with such a small sample size. Adjusting for covariates also has little chance of resulting in significance. The authors would do well to be more descriptive in presenting results. Although I appreciate the fact that the authors have used repeated measures analyses, they should acknowledge that given the small sample size, these would be unlikely to have significant results. Conclusions should not be based on not finding significant differences between groups.

• DEX-C: post-hoc analyses are not really appropriate here and should be excluded.

Discussion

• Discussion of feasibility results should be given precedence over “preliminary efficacy” in the discussion section. The authors, in discussing feasibility, should address the obvious fact that while at-home, highly individualized programs are more likely to be successful than not, they are also expensive and difficult to implement on a large scale. They are expensive in terms of the time of those administering the tests/services and conducting these in the home also requires scheduling effort and additional traveling time. This seems like the biggest drawback to this type of intervention and should definitely be discussed. Would insurance pay for this type of intervention? Because of the progressive nature of PD, it would have to be ongoing, and not a one-time intervention, which is also a concern in terms of cost.

• “Patients with longer disease duration registered improvement while their DRS-II scores were inferior to those with shorter disease duration, suggesting they benefited from interventions despite a more global cognitive impairment.” This is very likely due to regression to the mean.

• The assumption that patients if the psychoeducation/mindfulness group were motivated by their intervention to practice mindfulness between sessions is just an assumption. Involvement in a trial is also well-known to influence the behavior of the “control” group to adopt “intervention” practices.

• “…this sample size is typical for this kind of phase 1 study in the field of cognitive intervention.” Please omit this statement. I do not feel that doing something because “everyone else does it” is a valid justification. The reasons the authors have given sufficiently address this point.

• Again, the labor-intensive/expensive nature of the intervention should be addressed as a limitation.

Overall, this study has been nicely done and was a pleasure to review. I do feel that the authors should be more conservative in their conclusions, however, since the study was not powered for anything other than preliminary efficacy. It nonetheless offers important information on an intervention that has promise and in an area where it is widely needed. But more emphasis needs to be placed on feasibility of implementation than on efficacy, since the study is clearly underpowered to detect differences between the two groups, and a finding of “no difference” doesn’t mean there is no difference.

Reviewer #3: thanks alot for effort in this paper

i have minor issues want to be clear

Introduction

need editing and rewrite

Abstract:

Methods section is poorly framed. It has to be re-written.

Demographic profile of patients is not mentioned clear

Discussion:

needs to be as per well defined objectives

Describe sources of potential bias and imprecision

It has to be framed in such a way that readers are able to have good understanding of the current evidences and rationale of the paper

6. PLOS authors have the option to publish the peer review history of their article (what does this mean?). If published, this will include your full peer review and any attached files.

Reviewer #1: **Yes: **Gopal Nambi

Reviewer #2: No

Reviewer #3: No

---

## [Author Response · Author response to Decision Letter 0]

2 Nov 2021

Editorial comments

1. Thank you for updating your Data Availability statement: "Data availability: The data that support the findings of this study are available from the corresponding author, AGR, upon reasonable request.

Giguère-Rancourt, Ariane, 2020, "Cognitive Training in Parkinson's disease", https://can01.safelinks.protection.outlook.com/?url=https%3A%2F%2Fdoi.org%2F10.5683%2FSP2%2F0LMBRX&data=04%7C01%7Cariane.giguere-rancourt.1%40ulaval.ca%7C76e25e3f2895440eafdc08d98df6630e%7C56778bd56a3f4bd3a26593163e4d5bfe%7C1%7C0%7C637696911624743123%7CUnknown%7CTWFpbGZsb3d8eyJWIjoiMC4wLjAwMDAiLCJQIjoiV2luMzIiLCJBTiI6Ik1haWwiLCJXVCI6Mn0%3D%7C1000&sdata=1xMLTa1Y1kFvUt2%2Fsk9y%2B%2Box0UKjZEFTfnVSHimn3nU%3D&reserved=0, Scholars Portal Dataverse, V1, UNF:6:RAQT1DXYblrlo+gpYplHyA== [fileUNF]"

Please note that PLOS ONE's data policy requires a non-author institutional contact for data access in the interest of maintaining long-term data accessibility. At this time, please provide a non-author point of contact that is able to receive queries regarding data access.

RESPONSE: We added an institutional point of contact e-mail address for the database: gdr@ulaval.ca (the address of the research data manager at Laval university).

---

## [Decision Letter · Decision Letter 1]

29 Nov 2021

PONE-D-21-08484R1Goal Management Training and Psychoeducation / Mindfulness for Treatment of Executive Dysfunction in Parkinson’s disease: A Feasibility Pilot Trial.PLOS ONE

Dear Dr. Giguère-Rancourt,

Thank you for submitting your manuscript to PLOS ONE. After careful consideration, we feel that it has merit but does not fully meet PLOS ONE’s publication criteria as it currently stands. Therefore, we invite you to submit a revised version of the manuscript that addresses the points raised during the review process.

We look forward to receiving your revised manuscript.

Kind regards,

Walid Kamal Abdelbasset, Ph.D.

Academic Editor

PLOS ONE

Reviewers' comments:

Reviewer's Responses to Questions

**Comments to the Author**

1. If the authors have adequately addressed your comments raised in a previous round of review and you feel that this manuscript is now acceptable for publication, you may indicate that here to bypass the “Comments to the Author” section, enter your conflict of interest statement in the “Confidential to Editor” section, and submit your "Accept" recommendation.

Reviewer #1: All comments have been addressed

Reviewer #3: All comments have been addressed

2. Is the manuscript technically sound, and do the data support the conclusions?

Reviewer #1: Yes

Reviewer #3: Yes

3. Has the statistical analysis been performed appropriately and rigorously? 

Reviewer #1: Yes

Reviewer #3: Yes

4. Have the authors made all data underlying the findings in their manuscript fully available?

Reviewer #1: Yes

Reviewer #3: Yes

5. Is the manuscript presented in an intelligible fashion and written in standard English?

Reviewer #1: Yes

Reviewer #3: Yes

6. Review Comments to the Author

Reviewer #1: Reviewer comments

Thank you for giving me this opportunity to review this article.

Abstract:

1. Summarize the abstract (follow the abstract guidelines).

2. Include the study duration and eligibility criteria of study participants.

3. Mention the reports with 95%CI (Upper – lower limit) for all the variables.

4. The conclusion should be more concise and drawn on the basis of study reports.

Manuscript

1. Summarize the introductory part.

2. How come this trial is differing from reference number 9? – please justify.

3. Please describe about Psych – mind treatment and its benefits.

4. The authors fail to find and report the research gap in this session.

5. Include the clinical significance of this trial over clinicians, patients and researchers.

6. Present the manuscript as per CONSORT guidelines.

7. Include the study setting and study duration.

8. Include the reliability and validity of all the outcome measures used in the study.

9. Include the detail description of intervention and control group.

10. Include the method of sample size calculation with suitable reference.

11. Mention about the demographic details of the participants in the results section.

12. In the results section, please discuss about the treatment compliance rate, adverse effects and the number of dropouts.

13. Mention the reports with 95%CI (Upper – lower limit) for all the variables.

14. Report the effect size and MCID values of all the primary and secondary variables.

15. Summarize the discussion part and include the mechanism of interventions on different variables with recent references.

16. The conclusion should be more concise and drawn on the basis of study reports.

Reviewer #3: thanks alot for your response

7. PLOS authors have the option to publish the peer review history of their article (what does this mean?). If published, this will include your full peer review and any attached files.

Reviewer #1: **Yes: **Dr.Gopal Nambi

Reviewer #3: No

---

## [Author Response · Author response to Decision Letter 1]

21 Dec 2021

Referee(s)' Comments to Author:

Reviewer #1: Reviewer comments

Thank you for giving me this opportunity to review this article.

Abstract:

1. Summarize the abstract (follow the abstract guidelines). RESPONSE: the abstract was summarized in 300 words. 

2. Include the study duration and eligibility criteria of study participants. RESPONSE: Added. Please see the method section of the abstract.

3. Mention the reports with 95%CI (Upper – lower limit) for all the variables. RESPONSE: Added.

4. The conclusion should be more concise and drawn on the basis of study reports. RESPONSE: The conclusion was summarized.

Manuscript

1. Summarize the introductory part. RESPONSE: The introduction was briefly summarized.

2. How come this trial is differing from reference number 9? – please justify. RESPONSE: Couture et al. (2019) is a systematic review of all cognitive intervention RCTs for PD patients, whereas the present study is a feasibility pilot trial assessing the effects of two interventions that were never tested before with PD-MCI patients. 

3. Please describe about Psych – mind treatment and its benefits. RESPONSE: Please see the description provided at page 8-9.

4. The authors fail to find and report the research gap in this session. RESPONSE: Please see the last paragraph of the introduction, page 6, which tries to explain the research gap.

5. Include the clinical significance of this trial over clinicians, patients and researchers. RESPONSE: Please see the end of introduction at pages 5 and 6.

6. Present the manuscript as per CONSORT guidelines. RESPONSE: You will find at the end of the present letter the CONSORT Table. 

7. Include the study setting and study duration. RESPONSE: Study setting, and duration is described at page 8 and in figure 2.

8. Include the reliability and validity of all the outcome measures used in the study. RESPONSE: Please see page 14-15 for reliability and validity of outcome measures. 

9. Include the detail description of intervention and control group. RESPONSE: There was no control group, since this is a single blind randomized between group comparative study. Both groups are described in detail in the method section, please see page 8 and 9.

10. Include the method of sample size calculation with suitable reference. RESPONSE: The power analysis was withdrawn at the first round of revision, please see your comment 17 of the first letter: « What is the need of doing sample size calculation as it is a feasibility pilot trial? ». The power calculation was therefore deleted.

11. Mention about the demographic details of the participants in the results section. RESPONSE: Please see table 1 at page 11-12 and the first paragraph of the result section, page 15.

12. In the results section, please discuss about the treatment compliance rate, adverse effects and the number of dropouts. RESPONSE: Please see the feasibility results provided at pages 15, 16 and 17, as well as the discussion, page 21.

13. Mention the reports with 95%CI (Upper – lower limit) for all the variables. RESPONSE: Added in the text, please see the result section, pages 19-20.

14. Report the effect size and MCID values of all the primary and secondary variables. RESPONSE: Added in Table 2.

15. Summarize the discussion part and include the mechanism of interventions on different variables with recent references. RESPONSE: We basically restructured and re-written the entire text in the

Introduction and Discussion sections and discussed potential mechanism of interventions (please see page 26), as well as discussion with recent references (pages 23-24). 

16. The conclusion should be more concise and drawn on the basis of study reports. RESPONSE: Please see page 29 for the revised conclusion.

---

## [Decision Letter · Decision Letter 2]

13 Jan 2022

Goal Management Training and Psychoeducation / Mindfulness for Treatment of Executive Dysfunction in Parkinson’s disease: A Feasibility Pilot Trial.

PONE-D-21-08484R2

Dear Dr. Giguère-Rancourt,

We’re pleased to inform you that your manuscript has been judged scientifically suitable for publication and will be formally accepted for publication once it meets all outstanding technical requirements.

Kind regards,

Walid Kamal Abdelbasset, Ph.D.

Academic Editor

PLOS ONE

Additional Editor Comments (optional):

Reviewers' comments:

Reviewer's Responses to Questions

**Comments to the Author**

1. If the authors have adequately addressed your comments raised in a previous round of review and you feel that this manuscript is now acceptable for publication, you may indicate that here to bypass the “Comments to the Author” section, enter your conflict of interest statement in the “Confidential to Editor” section, and submit your "Accept" recommendation.

Reviewer #1: All comments have been addressed

Reviewer #3: All comments have been addressed

2. Is the manuscript technically sound, and do the data support the conclusions?

Reviewer #1: Yes

Reviewer #3: Yes

3. Has the statistical analysis been performed appropriately and rigorously? 

Reviewer #1: Yes

Reviewer #3: No

4. Have the authors made all data underlying the findings in their manuscript fully available?

Reviewer #1: Yes

Reviewer #3: Yes

5. Is the manuscript presented in an intelligible fashion and written in standard English?

Reviewer #1: Yes

Reviewer #3: Yes

6. Review Comments to the Author

Reviewer #1: Dear authors,

Really appreciate your efforts for satisfactorily addressed my comments.

Really appreciate your efforts for satisfactorily addressed my comments.

Really appreciate your efforts for satisfactorily addressed my comments.

Reviewer #3: thanks alot for your respose

7. PLOS authors have the option to publish the peer review history of their article (what does this mean?). If published, this will include your full peer review and any attached files.

Reviewer #1: **Yes: **Gopal Nambi

Reviewer #3: No

---

## [Editor Report · Acceptance letter]

9 Feb 2022

PONE-D-21-08484R2 

Goal Management Training and Psychoeducation / Mindfulness for Treatment of Executive Dysfunction in Parkinson’s disease: A Feasibility Pilot Trial. 

Dear Dr. Giguère-Rancourt:

I'm pleased to inform you that your manuscript has been deemed suitable for publication in PLOS ONE. Congratulations! Your manuscript is now with our production department. 

Kind regards, 

on behalf of

Dr. Walid Kamal Abdelbasset 

Academic Editor

PLOS ONE